The first juvenile specimens of Plateosaurus engelhardti from Frick, Switzerland: isolated neural arches and their implications for developmental plasticity in a basal sauropodomorph

Hofmann Rebecca rebecca_hofmann@ymail.com
Sander P. Martin
Division of Paleontology, Steinmann Institute, University of Bonn , Bonn , Germany
Anquetin Jérémy
Electronic publication date: 2014 Jul 3
Publication date: 2014
Volume: 2
Electronic Location ID: e458
Received 2014 Apr 1; Accepted 2014 Jun 6
Copyright: © 2014 Hofmann and Sander
Copyright year: 2014
Copyright holder: Hofmann and Sander
License: This is an open access article distributed under the terms of the Creative Commons Attribution License, which permits unrestricted use, distribution, reproduction and adaptation in any medium and for any purpose provided that it is properly attributed. For attribution, the original author(s), title, publication source (PeerJ) and either DOI or URL of the article must be cited.
License URL: https://creativecommons.org/licenses/by/4.0/

Keywords: Late Triassic, Norian, Switzerland, Basal Sauropodomorpha, Plateosaurus engelhardti, Juvenile, Neurocentral suture closure, Bone histology

Funding: Deutsche Forschungsgemeinschaft SA469/24-1 The research was funded by a grant (SA469/24-1) of the Deutsche Forschungsgemeinschaft. The funders had no role in study design, data collection and analysis, decision to publish, or preparation of the manuscript.

==============================
The dinosaur Plateosaurus engelhardti is the most abundant dinosaur in the Late Triassic of Europe and the best known basal sauropodomorph. Plateosaurus engelhardti was one of the first sauropodomorph dinosaurs to display a large body size. Remains can be found in the Norian stage of the Late Triassic in over 40 localities in Central Europe (France, Germany, and Switzerland) and in Greenland. Since the first discovery of P. engelhardti no juvenile specimens of this species had been described in detail. Here we describe the first remains of juvenile individuals, isolated cervical and dorsal neural arches from Switzerland. These were separated postmortem from their respective centra because of unfused neurocentral sutures. However the specimens share the same neural arch morphology found in adults. Morphometric analysis suggests body lengths of the juvenile individuals that is greater than those of most adult specimens. This supports the hypothesis of developmental plasticity in Plateosaurus engelhardti that previously had been based on histological data only. Alternative hypotheses for explaining the poor correlation between ontogenetic stage and size in this taxon are multiple species or sexual morphs with little morphological variance or time-averaging of individuals from populations differing in body size.

Introduction

The basal Sauropodomorpha are a presumably paraphyletic assemblage (Yates, 2003a; Yates & Kitching, 2003; Yates, 2004; Upchurch, Barrett & Galton, 2007) and form successive sistergroups to the largest terrestrial animals ever known, the Sauropoda, with which they form the Sauropodomorpha (von Huene, 1932). Basal sauropodomorphs were the dominant high-browsing herbivores from the late Norian until the end of the Early Jurassic, when they were replaced in dominance by sauropods (Barrett & Upchurch, 2005). The basal sauropodomorph Plateosaurus was one of the first larger-bodied dinosaurs. The first fossil remains of this taxon were found in 1834 at Heroldsberg near Nuremberg by Johann Friedrich Philipp Engelhardt. The first to describe the material was Herman von Meyer in 1837 naming it Plateosaurus engelhardti (Moser, 2003).

Basal Sauropodomorpha are important for understanding the unique gigantism of sauropod dinosaurs (Sander et al., 2004; Sander & Klein, 2005; Upchurch, Barrett & Galton, 2007; Cerda, Pol & Chinsamy, 2013) because they inform us about the plesiomorphic condition from which sauropod gigantism evolved. One such plesiomorphic condition may be the developmental plasticity seemingly present in Plateosaurus engelhardti, expressed in a poor correlation of ontogenetic stage and size (Sander & Klein, 2005). Developmental plasticity was initially hypothesized based on long bone histology (Sander & Klein, 2005), but in this paper we corroborate its presence based on body size at neurocentral suture closure, as documented by the first juvenile remains of P. engelhardti from Frick, Switzerland. Apart from the Plateosaurus bonebed in Frick, juveniles of Plateosaurus had already been mentioned from Greenland (Jenkins et al., 1994), but a detailed description of this material was not published.

Systematics of Plateosaurus

A premise of any hypothesis of developmental plasticity is that the sample in question is derived from a single species. This necessitates a review of the systematics of Plateosaurus. The remains of Plateosaurus occur in the middle to the late Norian of Germany (von Huene, 1926; von Huene, 1932; Galton, 2001), France (Weishampel, 1984), Switzerland (Sander, 1992), and Greenland (Jenkins et al., 1994). The type species of Plateosaurus is P. engelhardti (von Meyer, 1837). Several more species have been described from other localities in Germany such as P. trossingensis (Fraas, 1913) from Trossingen and P. longiceps (Jaekel, 1914) from Halberstadt, and P. gracilis (von Huene, 1908) from the Löwenstein Formation of Stuttgart. Currently the Plateosaurus finds from Halberstadt, Trossingen and Frick are assigned to one species: P. engelhardti. However, nomenclatorial controversy still surrounds this name (Galton, 1984a; Galton, 1984b; Galton, 1985a; Galton, 1985b; Galton & Bakker, 1985; Weishampel & Chapman, 1990; Galton, 1997; Galton, 1999; Galton, 2000; Galton, 2001; Moser, 2003; Yates, 2003b; Prieto-Márquez & Norell, 2011; Galton, 2012).

A massive abundance of Plateosaurus material found in Plateosaurus bonebeds (Sander, 1992) can be found at three localities: Halberstadt (Central Germany), Trossingen (Southern Germany) and Frick (Switzerland) (see Fig. 1). The locality in Switzerland is in an active clay quarry, the Gruhalde quarry, the Keller AG in Frick (Canton Aargau, Switzerland), where the first dinosaur fossils were discovered in 1963.

Figure 1 Plateosaurus localities in Germany, France and Switzerland.

The map displays localities of Plateosaurus remains found in Germany, France and Switzerland, modified after Moser (2003) and Weishampel & Westphal (1986). The Plateosaurus bonebeds (sensu Sander, 1992) are Halberstadt (Central Germany), Trossingen (Southern Germany) and Frick (Switzerland).

Plateosaurus from Frick: geological setting

Since the focus of this study lies on recently discovered juvenile Plateosaurus material, a review of this and other Plateosaurus bonebeds is necessary. The Gruhalde quarry exposes a section representing 20 million years of geological time, from the entire Middle Keuper (Upper Triassic) up to the late Sinemurian Obtusus clays (Early Jurassic) (Sander, 1990). The Middle Keuper sediments are about 20 m thick, the upper 19 m of this section are the Upper Variegated Marls (Rieber, 1985; Sander, 1992). Plateosaurus remains are embedded in the Upper Variegated Marls (Norian), which is partially equivalent in stratigraphy, lithology and clay mineralogy to the Knollenmergel and Feuerletten beds in Southern and Central Germany (Finckh, 1912; Matter et al., 1988), and Eastern France (Weishampel & Westphal, 1986). The Upper Variegated Marls at Frick mainly consist of reddish, grayish or greenish marls commonly containing carbonate concretions or layers (Sander & Klein, 2005). There are three horizons producing dinosaur remains (B Pabst, pers. comm., 2012), the lowermost of which represents the Plateosaurus bonebed and was the subject of the study by Sander (1992). The lowermost horizon is also the source of the material sampled histologically (Sander & Klein, 2005) and of the juvenile material described in this study.

The miring hypothesis of Plateosaurus bonebed origin

Mass accumulations of basal sauropodomorph remains in Frick, but also Halberstadt and Trossingen in Germany, share the same taphonomy, resulting in their description as Plateosaurus bonebeds (Sander, 1992). The sediments encasing the bones in all three localities are alluvial mudstones overprinted by pedogenesis, representing a floodplain in a semiarid climate. Apparently, Plateosaurus individuals as the heaviest animals in the environment were preferentially mired in shallow depressions when the mud was wet, acting as a deadly trap. Once the animal got stuck in the soft ground, trying to pull itself out, the mud liquified and the resulting undertow made it impossible to get out. This process happened several times over a long period, explaining the mass accumulations (Sander, 1992), which cannot be shown to represent mass death events, however. Sander (1992) noted the lack of animals of less than 5 m total body length and of juveniles in all Plateosaurus bonebeds. He suggested that this lack was due to smaller body size and the resultant negative scaling of the load on the juvenile feet, reducing the risk for animals of less than 5 m in body length to become mired. The miring hypothesis of Sander (1992) predicted that no juveniles would be found in Plateosaurus bonebeds. Until 2010 this prediction was not violated, although the discovery by Sander & Klein (2005) of developmental plasticity opened up the possibility that juveniles exceeding 5 m in body length would be found.

Nevertheless, it came as a surprise that juvenile remains of Plateosaurus were found in the locality Frick in 2010 and particularly 2011. The 2011 material represents the remains of at least two individuals and primarily consists of isolated neural arches found in a bone field catalogued as MSF 11.3. in the lowermost bone layer. The term ‘bone field’ had been introduced by Sander (1992).

Ontogenetic studies of sauropodomorph dinosaurs: bone histology and suture closure

In general, there are only two methods to ascertain the postnatal ontogenetic stage in a dinosaur individual: bone histology and suture closure patterns, particularly in the skull and the vertebral column.

The long bone histology of Plateosaurus engelhardti from the localities of Trossingen and Frick has been studied in more detail (Sander & Klein, 2005; Klein & Sander, 2007) than any other basal sauropodomorph including Massospondylus carinatus (Chinsamy, 1993; Chinsamy-Turan, 2005). The primary bone of Plateosaurus engelhardti consists of fibrolamellar bone tissue, indicating fast growth, but also reveals growth cycles demarcated by LAGs (lines of arrested growth). The histological sample (Sander & Klein, 2005; Klein & Sander, 2007) included individuals that were not fully grown, but specimens showing morphological indicators of skeletal immaturity were lacking at the time of these histological studies. More importantly, the histological ontogenetic stage of similar sized individuals shows great variation (Sander & Klein, 2005), indicating a poor correlation between body size and age, suggesting developmental plasticity (Sander & Klein, 2005) with growth probably being influenced by environmental factors. The basal sauropodomorphs Massospondylus carinatus and Mussaurus patagonicus do not seem to show such plasticity (Chinsamy, 1993; Chinsamy-Turan, 2005; Cerda, Pol & Chinsamy, 2013).

On the other hand, the histology of sauropod long bones received a great deal of attention (Curry, 1999; Sander, 1999; Sander, 2000; Sander & Tückmantel, 2003; Curry & Erickson, 2005; Sander et al., 2006; Klein & Sander, 2008; Sander et al., 2011). Sauropods revealed a fast-growing bone tissue, described as laminar fibrolamellar bone and a generally uniform histology. They grew along a genetically determined growth trajectory with a certain final size. Sauropods display a good correlation between body size/ontogentic stage and age with little individual variation in rate of growth and final size (Sander, 2000; de Ricqlès, Padian & Horner, 2003; Sander & Klein, 2005; Sander et al., 2011).

Yet another vertebral feature is important to determine osteological maturity: the stage of closure in the neurocentral suture between the neural arch and its centrum. Brochu (1996) observed different maturity stages in extant and extinct crocodilians by studying the neurocentral suture closure as an size-independent maturity criterion. He pointed out the presence of three different stages of neurocentral suture closure: open, partially closed and completely closed. The pattern of neurocentral suture closure plays another important role. In different groups and also within the same group different patterns of closure can be found.

Within Sauropodomorpha basal sauropodomorphs like Thecodontosaurus caducus (Yates, 2003a) and Unaysaurus tolentinoi (Leal et al., 2004) seem to show a pattern consistent with a posterior-anterior pattern of suture closure. Unfortunately the study on a close relative to Plateosaurus, Massospondylus carinatus, does not give a reliable pattern of a neurocentral suture closure due to the incompleteness of the material of also different specimens (Cooper, 1981). Recently described material of a juvenile basal sauropodomorph Yunnanosaurus robustus (Sekiya et al., 2013) indicates a roughly posterior-anterior pattern of suture closure. Within Sauropoda diverse patterns of suture closure can be recognized with suture closure spreading from more than one vertebral position and in some cases with no visible order (Ikejiri, 2003; Ikejiri, Tidwell & Trexler, 2005; Gallina, 2011).

Objectives of study

The current study has the major objective to describe and compare juvenile neural arch morphology of Plateosaurus. In addition, we want to evaluate the implications of the finds of juveniles for the miring hypothesis and for the hypothesis of developmental plasticity. We first give a detailed morphological description of neural arch morphology (laminae and fossae) of the immature isolated neural arches found in bone field 11.3. and compare it with the neural arch morphology of osteologically mature specimens of Plateosaurus. To address developmental plasticity, we need to determine individual body size. Since no femora from bone field 11.3. can be reliably associated with the isolated neural arches, morphometric analysis of the neural arches was used to calculate femur length of the juveniles as a proxy of body size. Femur length of the juveniles was then added to the Frick dataset on which the hypothesis of developmental plasticity was based. We thus tested if developmental plasticity is also reflected by the morphology of Plateosaurus engelhardti and not only in its histology. Finally, we evaluate the implications of the finds of juveniles for the miring hypothesis.

Material and Methods

Material

The juvenile specimens of Plateosaurus were excavated in the Gruhalde clay quarry of the Tonwerke Keller AG in Frick (Switzerland) as part of a bone field in 2011. The discovery was part of systematic paleontological excavations preceding clay mining each year since 2004. Already in 2010, a seemingly juvenile individual had been discovered but this specimen remains unprepared. Since the bone field yielding the 2011 juveniles was the third bone concentration encountered in the 2011 field season, the bones received consecutive collection numbers starting with “MSF 11.3.”. The site was destroyed by mining but the exact position of the bonefield was recorded (Swiss State Coordinates: 642 953.5/261 961, lowermost bone layer, 80–90 cm above base of the gray beds). Bone field 11.3. yielded several different juvenile bones besides the studied juvenile neural arches, namely isolated centra. These were not used for further analyses because they lack diagnostic features, making a reliable determination of the position within the vertebral column impossible. Vertebrae belonging to the caudal vertebral series were not included in this study because the neurocentral sutures were closed in all specimens. In addition, tail vertebrae can only be assigned to a general region in the tail and not to an exact position. Caudal vertebrae, however, will be considered in terms of morphological change during ontogeny later on.

The girdle skeleton of the juvenile individuals is represented by a right scapula, right coracoid, a right pubis, a left ischium, and the appendicular skeleton is represented by a left femur, a tibia, a fibula, a left humerus, and a radius. These bones probably derive from immature individuals since the length of the bones is much smaller than in adults. The articular surfaces at proximal and distal ends of appendicular bones still show an immature stage of ossification. A host of ribs and haemapophyses may also derive from the juveniles. This study focuses on the isolated neural arches from bone field 11.3. The sample includes 17 specimens of isolated neural arches belonging to the cervical and dorsal vertebral series (Table 1).

Table 1 List of juvenile neural arches from bone field 11.3 with their respective position determined.

The presacral vertebral column of Plateosaurus engelhardti consists of 10 cervical vertebrae (Axis to C10) and 15 dorsal vertebra (D1–D15). Positions D3, D5, D6 and D10/D11 can be recognized twice in the sample. Specimen MSF 11.3.348 is the only caudal vertebra to be studied since caudal neural arches, at least in the region, do not reveal characters to make a determination of its position possible. Specimens MSF 11.3.388 and MSF 11.3.169 were not assignable to a position due to poor preservation.

Specimen number	Position in vertebral column	
MSF 11.3.317	Axis	
MSF 11.3.258	C3	
MSF 11.3.371	C4	
MSF 11.3.074	C6	
MSF 11.3.366	C10	
MSF 11.3.388	C?	
MSF 11.3.360	D3	
MSF 11.3.376	D3	
MSF 11.3.049	D4	
MSF 11.3.067	D5	
MSF 11.3.167	D5	
MSF 11.3.095	D6	
MSF 11.3.107	D6	
MSF 11.3.339	D7	
MSF 11.3.241	D10/D11	
MSF 11.3.303	D10/D11	
MSF 11.3.169	D?	
MSF 11.3.348	Cd?	

During the excavation, bone field 11.3. was covered with transparent foil to document the position of the bones found. This map shows that all bones were distributed over the whole area with no recognizably articulation or connection to each other (Fig. S1). The next step was to ascertain how many animals are represented and if specimens of different ontogenetic stage are recognizable. There are at least one adult and two juvenile animals represented by bonefield 11.3.

Assignment of the juveniles to P. engelhardti

Plateosaurus specimen SMNS 13200 from Trossingen (Fraas, 1913; von Huene, 1926; Yates, 2003b) is the standard for the morphological description of the juvenile specimens. The plateosaurs from Trossingen are assigned to P. engelhardti (Galton, 1997; Galton, 2000; Galton, 2001; Galton, 2012). The neural arch morphology of the Frick material described in this study is consistent with the morphology found in SMNS 13200. All of the Plateosaurus specimens found in Frick show the same variance related to sexual dimorphism, intraspecific variation and the final size found in the Trossingen specimens (Sander & Klein, 2005; Klein & Sander, 2007).

Preservation of the neural arches

The preservation of the bones in bone field 11.3. is characterized by various degrees and directions of diagenetic compaction, making the description of the neural arches sometimes challenging. The preservation ranges from no obvious compaction to slight dorsolateral distortion and heavily dorsolateral distortion with three dimensional preservation (see description of neural arches). The most obvious feature are fractures, going through the bones. Some of the neural arches and other bones like the ischium show another new feature, which has not been seen before in material from Frick: tectonic cracks originating from regional deformation, which are filled in with a ferrous mineral during diagenetic processes (see Figs. 2D, 2E, 4A, 5A, 9A–9C, 10B, 10C, 11A–11C, 13B, 13C and 18).

Figure 2 Specimen MSF 11.3.317 (axis) and MSF 11.3.258 (C3).

Plateosaurus engelhardti from Frick, Switzerland. Anterior neural arches of late juveniles. A–C: MSF 11.3.317 (axis) in A, left lateral view; B, dorsal view and C, ventral view. Specimen MSF 11.3.317 shows prezygapophyses facets being smaller and shorter than those of the postzygapophyses. The spof is the only fossa developed. D–F: MSF 11.3.258 (C3) in D, dorsal view; E, ventral view and F, right lateral view. The spof in MSF 11.3.258 gets deeper and the sprf is developed. Note the zipper-like structures on the pedicels in ventral view. Furthermore the specimen shows tectonic cracks in dorsal view. See text for abbrevations. Scale bars measure 1 cm.

Figure 3 Specimen MSF 11.3.371 (C4).

Plateosaurus engelhardti from Frick, Switzerland. Anterior cervical neural arch of a late juvenile. A–E: MSF 11.3.371 (C4) in A, left lateral view; B, dorsal view; C, ventral view; D, anterior view and E, posterior view. A partly preserved diapophysis on the right lateral side is visible. See text for abbreviations. Scale bars measure 1 cm.

Figure 4 Specimen MSF 11.3.074 (C6).

Plateosaurus engelhardti from Frick, Switzerland. Posterior cervical neural arch of a late juvenile. A–B: MSF 11.3.074 (C6) in A, right lateral view and D, dorsal view. Articular facets of the prezygapophyses and postzygapophyses are rough, suggesting a cover by cartilage. The diapophyses are well developed. Specimen MSF 11.3.074 shows tectonic cracks in right lateral view. See text for abbreviations. Scale bars measure 1 cm.

Figure 5 Specimen MSF 11.3.366 (C10).

Plateosaurus engelhardti from Frick, Switzerland. Posterior cervical neural arch of a late juvenile. A–B: MSF 11.3.366 (C10) in A, dorsal view and B, left lateral view. Specimen MSF 11.3.366 represents the cervicodorsal transition of posteriormost cervicals and anteriormost dorsals very well. Transverse processes (diapophyses and parapophyses) are changing in shape, size and function. Therefore all of the diapophyseal laminae and fossae are well developed. Tectonic cracks are present in left lateral view on the prezygapophyses and postzygapophyses on both sides of the neural arch. See text for abbreviations. Scale bars measure 1 cm.

Adult specimens studied for comparison

A morphological comparison of the studied juvenile specimens to other specimens, especially osteologically mature individuals, is important. This may reveal ontogenetic morphological variation (Carballido & Sander, 2013). We studied three Plateosaurus vertebral columns in detail for comparison. Two of these are from Frick (MSF 5, MSF 23) and one is from Trossingen (cast of SMNS 13200), see Fraas (1913), von Huene (1926), Galton (1985a), Galton (1986) and Sander (1992) on this material.

Specimen MSF 5 consists of a block with two incomplete individuals of Plateosaurus, with the smaller animal lying on the top of a larger one. The larger animal (MSF 5B) preserves the anterior half of the skeleton with a partial and partially disarticulated skull, articulated vertebrae from the first cervical to the fifth dorsal vertebra and several other disarticulated elements as shown in Fig. S2 (Rieber, 1985; Galton, 1986; Sander, 1992). The smaller individual (MSF 5A) is represented by a left humerus being smaller compared to the right humerus of the big animal lying on the right side of the block. The remains are prepared right-side up and still remain in the sediment (Sander, 1992). The specimen is exhibited in situ at the Sauriermuseum Aathal (SMA) on permanent loan from the MSF. For this study the complete cervical and partly preserved dorsal vertebrae series of MSF 5B is of interest. MSF 5B represents the most complete and best preserved articulated cervical and partial dorsal series from the second cervical vertebra (C2) to the fifth dorsal vertebra (D5) found in Frick, anterior body regions being underrepresented due to the specific taphonomy of the locality (Sander, 1992). The bones did not suffer much distortion compared to the bone field 11.3. specimens and are well preserved in three dimensions. All of the vertebrae of MSF 5B show completely closed neurocentral sutures. The neural arches show well and fully developed laminae and fossae throughout the vertebral series with no feature missing.

MSF 23 is a nearly complete and essentially articulated skeleton of a Plateosaurus from Frick, on display at the Sauriermuseum Frick (Sander, 1992) (Fig. S3). The morphology of the skeleton has not yet been described in detail, but it was figured by Sander (1992, Fig. 3) as well as in the non-technical literature (Sander, 1993; Sander, 2012). The segment of C2–C7 is articulated but separated by a fault from C8 to D15 that follow in full articulation. At a first glance, D15 may be considered to belong to the sacrum, being a dorsosacral, because it seems to be fused with the ilia on both sides. However no other Plateosaurus revealed more than three sacrals (Jaekel, 1914; Galton, 1999; Galton, 2001). The diapophyses of D15 also are not as massive in their morphology as those of the sacrals. So the adhesion of D15 to the anteriormost part of the sacrum may be due to the age of the animal. The neural arches in MSF 23 generally experienced strong dorsolateral pressure from the right side during diagenesis. This led to extreme deformation of the vertebrae in the specimen. Nevertheless MSF 23 shows fully developed vertebral morphology with all laminae and fossae being present. All of the neurocentral sutures are completely closed.

The third specimen is a cast of a complete skeleton from Trossingen (SMNS 13200, Fig. S4), exhibited at the Naturama (NAA) in Aarau (Switzerland). SMNS 13200 was excavated as a nearly complete articulated skeleton in 1911 in the Knollenmergel Beds of Trossingen at the Obere Mühle (Fraas, 1913) and forms the basis of the osteological description by von Huene (1926). The left forelimb distal to the humerus is missing, and the tail is incomplete as well, missing some vertebrae. However, the presacral vertebral column is complete and well preserved. SMNS 13200 shows good three dimensional preservation with no or little influence of compaction on the vertebral column. All vertebrae display well developed laminae and fossae with all neurocentral sutures being closed.

Methods

The morphological description of the neural arches follows the nomenclature of Wilson (1999) and Wilson et al. (2011). Because of their complex morphology and because morphological characters change sequentially throughout the axial skeleton (Carballido et al., 2012; Carballido & Sander, 2013), sauropodomorph neural arches can be assigned to specific positions in the vertebral column with a margin of error of one position or less. In sauropods, not only do vertebral characters change within one animal but also during the ontogeny of the same animal (Carballido et al., 2012). Before a sauropod reached osteological maturity, its vertebrae pass developmental stages, often displaying more primitive characters known in more basal taxa. To determine the stage of osteological maturity of the juvenile Plateosaurus a direct comparison of morphological characters to osteologically mature (completely closed neurocentral sutures) plateosaurs is necessary.

Terminology of laminae and fossae

The morphological description of the neural arches of this study follows the nomenclature of Wilson (1999) for the laminae and Wilson et al. (2011) for the fossae of sauropod dinosaurs which can be applied to basal sauropodomorphs as well (Wilson, 1999; Wilson et al., 2011). The nomenclature for laminae developed by Wilson (1999) is based on landmarks on the vertebra, namely the connections a lamina establishes, whereas the nomenclatures previously used by other scientists were mainly based on the origin the laminae have. The fossae’s names are defined by the surrounding laminae (Wilson et al., 2011). Anatomical abbreviations are listed at the end of the text.

Morphometrics and 3D visualization

Simple morphometric analysis was applied to estimate the body length of the juvenile Plateosaurus from measurement that can be taken on neural arches. In dinosaurs, femur length is a reliable proxy for body mass (Carrano, 2006). In the case of Plateosaurus, femur length equals approximately 1/10 of body length (Sander, 1992). Since our material only consists of isolated neural arches, we had to establish a new proxy which is suitable for determining the body lengths of the juveniles. We decided to use the zygapophyseal length of the neural arches for developing a proxy. Due to the extreme dorsolateral compression of some specimens and the better preservation of the pre- and postzygapophyses compared to the transerve processes of the neural arches, measuring zygapophyseal length appears to be the most reliable size proxy. The zygapophyseal length of the neural arches of all specimens studied was measured from the tip of the prezygapophysis to the tip of the postzygapophysis, thus representing the maximum anteroposterior length of a neural arch. For the calculations of body lengths of the juveniles, femur length of specimens MSF 5B, MSF 23 and SMNS 13200 was measured (Table 2) as maximal length on the medial side. The femur of specimen MSF 5B is not preserved, but its scapula is. Based on the scapula/femur ratio (76%) of specimen MSF 23 and on measured scapula length of MSF 5B, we were able to calculate the femur length of MSF 5B.

Table 2 Femur length of the adult specimens MSF 5B, MSF 23 and SMNS 13200.

The femur length of the adult specimens MSF 5B, MSF 23 and SMNS 13200 with completely closed neurocentral sutures on their vertebral column. The femur length of specimen MSF 5B was calculated with the given scapula/femur ratio (76%) of specimen MSF 23 and the measured scapula length of MSF 5B, since the femur itself is not preserved.

Adult (osteologically mature) specimens	Femur length (mm)	
MSF 5B	565	
MSF 23	610	
SMNS 13200	685	

The ratio between zygapophyseal length and femur length of MSF 5B, MSF 23, and SMNS 13200 were measured to calculate the femur lengths of the juvenile specimens of bone field 11.3. (Table S2). Our limited sample size of three adult specimens for comparison to the juvenile material is due to the taphonomy of Plateosaurus bonebeds, leading to incomplete and in most cases disarticulated finds. For the calculation of the femur length of the juveniles in percentage, we only used the data of specimen SMNS 13200, where the material is the most complete and best preserved, compared to all other specimens studied.

The main problems during measurements of zygapophyseal length in neural arches of all specimens studied were caused by poor preservation in some bones, with the tips of pre- or postzygapophyses missing. Sometimes heavy deformation, e.g., in MSF 23 in the region of the posteriormost dorsal vertebrae, made measurements impossible. In articulated specimens like MSF 5B and MSF 23, bones like dorsal ribs and gastralia obscure parts of the vertebral column.

Morphometric measurements were taken with a sliding caliper for distances between 0 and 150 mm. If the distance was greater than 150 mm, or the measurement was not accessible with the sliding caliper, a measuring tape was used. The measurements were taken to the nearest 0.1 mm (calliper) and to the nearest millimeter (measuring tape).

In terms of visualization of the isolated neural arches, we created three-dimensional digital models of four representative specimens (Files S1–S4). These specimens are two anterior cervical neural arches (MSF 11.3.317 and MSF 11.3.258), one middle dorsal neural arch (MSF 11.3.339), and the posteriormost dorsal neural arch (MSF 11.3.303). We computed these models with the photogrammetry software Agisoft PhotoScan using sets of professional-grade photographs of the specimens.

Results

Description

Among the juvenile bones, there are six isolated neural arches (specimens are listed in Table 1 and Supplemental Information) that can be assigned to the cervical vertebral column. This is based on their low and elongated appearance in comparison to the taller and shorter proportions of the dorsal neural arches (von Huene, 1926). We identified eleven dorsal neural arches from the bone field 11.3. sample (specimens are listed in Table 1 and Supplemental Information). The specimens can be further subdivided into anterior (C1–C5) and posterior (C6–C10) cervical neural arches and into anterior (D1–D5), middle (D6–D10) and posterior (D11–D15) dorsal neural arches. The identification of the position of the neural arches are performed with the help of characters and features of diapophyses (d), prezygapophyses (prz), postzygapophyses (poz), parapophyses (pa), and the neural spines, as described by von Huene (1926) and Bonaparte (1999). The laminae and fossae play an important role in the morphology of the neural arch (Bonaparte, 1999; Wilson, 1999; Wilson et al., 2011). Furthermore the processes of the neural arch change gradually along the vertebral column, e.g. in length, shape, size, location on the arch and angle at which these stand out from the vertebra (Wilson, 1999).

The complete vertebral column of Plateosaurus engelhardti consists of a rudimentary proatlas, 10 cervical vertebrae, 15 dorsal vertebrae, three sacral vertebrae, and at least 50 caudal vertebrae (von Huene, 1926; Bonaparte, 1999; Upchurch, Barrett & Galton, 2007). Specimens MSF 11.3.388 (cervical neural arch) and MSF 11.3.169 (dorsal neural arch) displayed the worst preservation and were not described in detail. We were unable to reliably determine the position of these two arches within the vertebral column since all of the important characters were not preserved.

Cervical neural arches

Axis, MSF 11.3.317 (Figs. 2A–2C, File S1)

The axis is the anteriormost neural arch identified in the bone field. With the diapophysis and parapophysis missing, the diapophyseal and parapophyseal laminae are not present in the axis. The prezygapophyses show much smaller and shorter facets than the postzygapophyses. The prezygapophysis is ventrally supported by a single cprl. The tprl connecting both prezygapophyses is missing. Short sprl’s line up dorsally to the neural spine. As a counterpart the cpol holds up the postzygaphysis, and the spol runs up dorsally from the postzygapophysis towards the neural spine. A poorly developed tpol connecting the postzygapophyses is present. The only fossa is the spof, but it does not extend deeply into the neural arch. In ventral view, the pedicels show the zipper-like surface of the neurocentral suture, which is typical for morphologically immature bones originating from the open neurocentral suture (Brochu, 1996; Irmis, 2007). Furthermore, the articular surfaces of the postzygapophyses in ventral view are rough and were only partly ossified at the time of death. The morphology of the axis arch does not differ from the adult.

Third cervical, MSF 11.3.258 (Figs. 2D–2F, File S2)

The neural arch can be assigned to the third position within the vertebral column. No diapophysis or parapophysis is present, therefore the arch is missing any diapophyseal and parapophyseal laminae. Postzygapophysis and prezygapophysis are both small and form a low angle, indicating that this neural arch is an anterior cervical one. The tprl (the connecting lamina between the prezygapophyses) and tpol are well developed. The sprl is hardly developed in contrast to the spol being quite present. The cprl and cpol are well developed. Like in the axis, the spof is present and becomes deeper. Though less developed, the sprf is present now. Zipper-like suture surfaces on the pedicels are recognizable in ventral view.

Fourth cervical, MSF 11.3.371 (Figs. 3A–3E)

The arch shows a partly preserved diapophysis on the right lateral side, but still no parapophysis is present. Nevertheless, diapophyseal as well as parapophyseal laminae do not extend onto the arch. The prezygapophyses of MSF 11.3.371 are much more elongated compared to prezygapophyses in more anterior neural arches and the postzygapophyses of the same arch. The articular surfaces have a quite low angle of less than 45°. While the cpol remains short in length, the cprl is a thick elongated lamina. Sprl and spol are well developed along with the sprf and the spof, with the spof being the deeper and broader fossa. Other fossae are not present.

Sixth cervical, MSF 11.3.074 (Figs. 4A and 4B)

The partly preserved diapophysis fully moved dorsally onto the neural arch and is situated at the midlength of the neural arch. No parapophysis is present. The prezygapophysis and postzygapophysis seem to be very steeply angled, and the surface of the articular facets is rough, suggesting a cover by cartilage. This is unlike in adults, where zygapophyseal articular facets are well ossified and smooth. Intense lateral compaction of the arch with a sligthly ventral to dorsal shift is recognizable. The acdl emerges as a thin lamina going anterodorsally up from the anterior part of the junction between centrum and neural arch to the tip of the prezygapophysis, recognizable on both the left and right lateral side; concomitant with the presence of a small and shallow prcdf. The pcdl is not present. The neural spine is higher than in the anterior cervicals. Sprl, spol, tprl and tpol are present. Both cprl and cpol seem to be shorter than in the more anterior cervical arches. The sprf is not well developed whereas the spof is deeper. The pedicels lack the zipper-like structures due to poor preservation.

Tenth cervical, MSF 11.3.366 (Figs. 5A and 5B)

In the tenth cervical neural arch the cervicodorsal transition is visible. Posteriormost cervical neural arches show strong reduction in centrum and zygapophyseal length in comparison to the previous arches. The neural spine gets higher. The size, shape and functions of diapophyses change due to dorsal ribs which have to be supported. The ribs are supported from below by parapophyses which migrate onto the dorsal neural arches (Wilson, 1999). Though the diapophysis of the specimen is not complete with the tip missing, the diapophysis arises fully from the neural arch. As a consequence, all of the diapophyseal laminae are present and well developed. These include the acdl, which is a thin lamina in the sixth cervical (MSF 11.3.074), but which is thickened and well established in specimen MSF 11.3.366. The diapophysis is well supported ventrally by the pcdl, being the stronger and broader lamina, and the acdl. The cdf is still simple and not deep. On the contrary, the prcdf and pocdf are deep and extensive. The surface of the prezygapophyses and postzygapophyses are much more extensive, which is not the case for zygapophyses of anterior cervicals. Still, a parapophysis is not visible, but the laminae connecting the diapophysis with the prezygapophysis (prdl) and the diapophysis with the postzygapophysis (podl) are distinctly developed. All the other laminae like sprl, spol, tprl, tpol, cprl and cpol as well as sprf and spof are well developed. In contrast to the neural spine of more anterior cervicals, the neural spine of this specimen is much thicker. Specimen MSF 11.3.366 is the anteriormost specimen in the cervical series to exhibit a hyposphene and hypantrum for further support of the vertebral column.

Dorsal neural arches

Anterior neural arches from the first to the seventh dorsal are most abundant in bone field 11.3., and only two posterior dorsal neural arches can be recognized. Some positions are represented twice like the third, the fifth, the sixth and the tenth/eleventh dorsal. All of the dorsal neural arches show well developed hyposphenes and hypantra if this region is preserved. Zipper-like sutural surfaces are preserved for the dorsals MSF 11.3.360, MSF 11.3.167, MSF 11.3.095, MSF 11.3.107 and MSF 11.3.339.

Third dorsal, MSF 11.3.360 (Figs. 6A–6D)

This specimen is one of the most anterior dorsal neural arch in the dorsal series. With the shortest and thickest neural spine within the whole vertebral series, being nearly square in shape in dorsal view and sticking out from the arch at a right angle, the neural arch can be identified as a third dorsal (von Huene, 1926). The diapophysis is slightly oblique and gently posteriorly directed. Furthermore, three very deep fossae are well recognizable below the diapophysis (prcdf, pocdf and cdf). A first sign of a slight parapophysis articular facet is recognizable on both sides of the bone. The parapophysis still seems to have been located more on the centrum than on the neural arch. The much broader facets of the prezygapophyses in comparison to small ones of the postzygapophyses are remarkable. Nonetheless, both show rough articular sufaces like all the cervical neural arches. All laminae (acdl, pcdl, prdl, podl, sprl, spol, cprl and cpol) are fully developed.

Figure 6 Specimen MSF 11.3.360 (D3).

Plateosaurus engelhardti from Frick, Switzerland. Anterior dorsal neural arch of a late juvenile. A–D: MSF 11.3.360 (D3) in A, right lateral view; B, dorsal view; C, anterior view and D, posterior view. Specimen MSF 11.3.360 has the shortest and thickest neural spine of all neural arches studied. Parapophysis articular facets slightly become visible. See text for abbreviations. Scale bars measure 1 cm.

Third dorsal, MSF 11.3.376 (Figs. 7A–7C)

Specimen MSF 11.3.376 can also be identified as a D3 due to the same diagnostic characters. However, there are some striking differences in comparison to the previous specimen. The prezygapophyses are much smaller and seem to be elongated instead of being broad. This may be due to preservation, though the shape of MSF 11.3.360 appears to be little affected by diagenetic deformation. MSF 11.3.376 experienced dorsoventral crushing. In addition, the parapophysis articular area has clearly developed and is situated on the neural arch while the parapophyses of MSF 11.3.360 still articulate with the centrum, because they are hardly visible. All laminae are fully present and developed, whereas the acdl is slightly truncated by the parapophysis articular facet.

Figure 7 Specimen MSF 11.3.376 (D3).

Plateosaurus engelhardti from Frick, Switzerland. Anterior dorsal neural arch of a late juvenile. A–C: MSF 11.3.376 (D3) in A, right lateral view; B, dorsal view and C, posterior view. This specimen shows the same diagnostic characters as specimen MSF 11.3.360 except for very well developed parapophysis articular facets. See text for abbreviations. Scale bars measure 1 cm.

Fourth dorsal, MSF 11.3.049 (Figs. 8A and 8B)

In the fourth dorsal neural arch, the thickness of the spine decreases a little and the spine gets longer. Unfortunately the tip of the diapophysis is missing on both sides. No parapophysis is visible. In all likelihood, the parapophysis articular facet is situated on the centrum. This may lead to the assumption that we deal with a cervical, but the neural spine indicates the specimen to be a dorsal. The appearance of the prezygapophyses and the very short postzygapophyses also argue for a dorsal neural arch. Fossae and accompanying laminae are well developed. All three fossae below the diapophysis are very deep and well visible (prcdf, cdf and pocdf). No parapophysis influences the laminae and fossae. Tough the cdf seems to be not as deep as in the third dorsals. The well established laminae and fossae indicate the neural arch belongs to a fourth dorsal. Cprl and cpol distinctly arise from the prezygapophysis and postzygapophysis, increasing the general height of the neural arch.

Figure 8 Specimen MSF 11.3.049 (D4).

Plateosaurus engelhardti from Frick, Switzerland. Anterior dorsal neural arch of a late juvenile. MSF 11.3.049 (D4) in A, right lateral view and B, dorsal view. No parapophyses are visible. All of the diapophyseal laminae and fossae are well developed. See text for abbreviations. Scale bars measure 1 cm.

Fifth dorsal, MSF 11.3.067 (Figs. 9A–9C)

The fifth dorsal neural arch shows partly preserved diapophyses, but no parapophysis articular facet due to poor preservation. The neural spine shows that a posterior inclination is seen from now on backwards in the vertebral column. The left lateral side of the arch shows that all the laminae and fossae are well developed in this specimen. As expected, the prcdf begins to diminish in size and extent due to the parapophysis articular facet moving dorsally onto the neural arch, also slowly closing the acdl, separating the lamina into acpl and ppdl in posterior dorsal neural arches. In addition, the parapophysis articular facet also influences the prdl to the extent that it recedes. This process takes place stepwise, visibly beginning in the fifth dorsal and being complete in the eighth dorsal in which there are only two fossae left below the diapophysis (pocdf and cdf).

Figure 9 Specimen MSF 11.3.067 (D5).

Plateosaurus engelhardti from Frick, Switzerland. Anterior dorsal neural arch of a late juvenile. MSF 11.3.067 (D5) in A, left lateral view; B, dorsal view and C, ventral view. All laminae and fossae are still well developed, but the prcdf begins to decrease in size and extent due to the parapophysis articular facets moving upwards onto the neural arch. Specimen MSF 11.3.067 shows tectonic cracks in dorsal view on the left lateral prezygapophysis and on the right lateral diapophysis. See text for abbreviations. Scale bars measure 1 cm.

Fifth dorsal, MSF 11.3.167 (Figs. 10A–10C)

This is another neural arch belonging to a fifth dorsal vertebra. The specimen is heavily crushed on the left side, leaving the right side for the description. All laminae are well developed beneath the diapophysis with deep fossae (pcdl, acdl, prdl, podl, sprl, spol, cprl and cpol). A parapophysis articular facet is present interrupting the acdl. The appearance of the zygapophyses conforms with those of specimen 11.3.067.

Figure 10 Specimen MSF 11.3.167 (D5).

Plateosaurus engelhardti from Frick, Switzerland. Anterior dorsal neural arch of a late juvenile. MSF 11.3.167 (D5) in A, right lateral view; B, dorsal view and C, ventral view. On the neural arch of this specimen the parapophysis articular facet is well visible. The cdf contains collapsed bone caused by crushing. Dessication cracks are present in dorsal view on the right lateral prezygapophysis and in ventral view on the pedicels on both sides. See text for abbreviations. Scale bars measure 1 cm.

Sixth dorsal, MSF 11.3.095 (Figs. 11A–11C)

Specimen MSF 11.3.095 is assigned to the sixth position in the dorsal vertebral column. The diapophyses are posteriorly oriented, suggesting a middle dorsal neural arch. The prezygapophyses are elongated in contrast to the postzygapophyses being shorter and smaller in expanse. Furthermore all laminae are fully developed. At the anterior end of the arch, dorsal of the neurocentral suture, a distinctive parapophysis articular facet is present on both sides. The parapophysis articular facet displaces the acdl, giving rise to the ppdl, connecting the parapophysis from ventral to dorsal with the diapophysis, and the acpl and the prpl. The prpl connects the parapophysis anterodorsally with the prezygapophysis. The prdl is still well visible. All the rest of the laminae are well developed, like in the arches described before. The same applies to all of the fossae. Further evidence for the identification of the specimen as a sixth dorsal is that the prcdf becomes narrower and decreases in depth compared to the prcdf in more anterior neural arches.

Figure 11 Specimen MSF 11.3.095 (D6).

Plateosaurus engelhardti from Frick, Switzerland. Middle dorsal neural arch of a late juvenile. MSF 11.3.095 (D6) in A, left lateral view; B, dorsal view and C, ventral view. The parapophysis articular facet displaces the acdl, giving rise to the ppdl, the acpl and prpl. Due to this change of laminae the prcdf becomes smaller in extent. Specimen MSF 11.3.095 shows tectonic cracks in left lateral view, dorsal view as well as ventral view. See text for abbreviations. Scale bars measure 1 cm.

Sixth dorsal, MSF 11.3.107 (Figs. 12A–12C)

This specimen can also be identified as a sixth dorsal neural arch. All features seen in this specimen coincide with those of specimen MSF 11.3.095. The bone is complete although the diapophysis is broken off on the left side and is diagenetically recemented to the arch.

Figure 12 Specimen MSF 11.3.107 (D6).

Plateosaurus engelhardti from Frick, Switzerland. Middle dorsal neural arch of a late juvenile. MSF 11.3.107 (D6) in A, right lateral view; B, ventral view and C, posterior view. All of the characters found coincide with those of MSF 11.3.095. See text for abbreviations. Scale bars measure 1 cm.

Seventh dorsal, MSF 11.3.339 (Figs. 13A–13C, File S3)

Although being the most complete and best preserved specimen of all, this neural arch is strongly influenced by anteroposterior compaction. This implies an extremely posteriorly directed diapophysis and a constrained elongation of the prdl on the right lateral side. Aside from the preservation, the prdl is much shorter and more inconspicuous than in the more anterior neural arches which argues for a position around the seventh dorsal, where the prdl is fused with the ppdl, the acdl is consumed by the acpl, and the cprl is disrupted by the prpl, connecting the parapophysis anterodorsally with the prezygapophysis. Unfortunately, no parapophysis articular facet is preserved. Furthermore, the specimen impressively shows the rough and only partly ossified zygapophyseal articular surfaces.

Figure 13 Specimen MSF 11.3.339 (D7).

Plateosaurus engelhardti from Frick, Switzerland. Middle dorsal neural arch of a late juvenile. MSF 11.3.339 (D7) in A, right lateral view; B, dorsal view and C, ventral view. The prcdf is extremely diminished in comparison to anterior dorsal neural arches. The articular surfaces of the prezygapophyses, postzygapophyses, and diapophyses display very rough articular surfaces once being covered by cartilage. The abrasion is an indicator of osteological immaturity. Specimen MSF 11.3.339 shows tectonic cracks in ventral view on the left lateral prezygapophysis. See text for abbreviations. Scale bars measure 1 cm.

Tenth/Eleventh dorsal, MSF 11.3.241 (Figs. 14A–14C)

This is the posteriormost position represented by the neural arches found in bone field 11.3., being the tenth or eleventh dorsal neural arch. The arch has broad and extensive diapophyses, oriented nearly at right angles to the arch. The partly preserved neural spine does not show any indication of a bifurcation in the posterior part, which is mainly the reason why the neural arch cannot be assigned to the 12th up to the 15th dorsal. A sure indicator for a posterior dorsal position are the presence of only two fossae below the diapophyses. The prdl has fully vanished from the arch in this position. A parapophysis articular facet is well preserved on the left lateral side of the specimen. Prezygapophyses and postzygapophyses are both short compared to prezygapophyses and postzygapophyses in the middle dorsal neural arches (i.e., the fifth, sixth, and seventh dorsal). In all middle and posterior dorsal neural arches, the articular surfaces of the zygapophyses are horizontal. At the same time, the hyposphene and hypantrum are very distinctive.

Figure 14 Specimen MSF 11.3.241 (D10/D11).

Plateosaurus engelhardti from Frick, Switzerland. Middle/posterior dorsal neural arch of a late juvenile. MSF 11.3.241 (D10/D11) in A, right lateral view; B, dorsal view and C, ventral view. The diapophyses are broad and extensive. Only two diapophyseal fossae are present (cdf and pocdf). Hyposphene and hypantrum are very distinctive. See text for abbreviations. Scale bars measure 1 cm.

Tenth/Eleventh dorsal, MSF 11.3.303 (Figs. 15A–15C, File S4)

This posterior dorsal neural arch can also be assigned to a position around the tenth dorsal. The diapophyses are not well preserved, missing the tip on the right lateral side and not being preserved on the left lateral side, to which a partly preserved bone (MSF 11.3.304) is cemented. Presumably this bone is a posterior caudal vertebra. Again the diapophysis is directed laterally at a 90-degree angle like in specimen MSF 11.3.241. The shape and appearance of the prezygapophyses and postzygapophyses also coincide with those of the previously described specimen. In contrast to specimen MSF 11.3.241, the postzygapophyses show completely ossified articular surfaces. All laminae and fossae are well developed.

Figure 15 Specimen MSF 11.3.303 (D10/D11).

Plateosaurus engelhardti from Frick, Switzerland. Middle/posterior dorsal neural arch of a late juvenile. MSF 11.3.303 (D10/D11) in A, right lateral view; B, dorsal view and C, ventral view. In ventral view a partly preserved posterior caudal vertebrae (MSF 11.3.304) is cemented to specimen MSF 11.3.303. See text for abbreviations. Scale bars measure 1 cm.

Minimal number of individuals (MNI)

The assignment to the position of the neural arches indicates the minimum number of juvenile individuals (MNI) represented in bone field 11.3. In the dorsal series, some positions are represented twice, such as the the third dorsal (MSF 11.3.360 and MSF 11.3.376), the fifth dorsal (MSF 11.3.067 and MSF 11.3.167), the sixth dorsal (MSF 11.3.095 and MSF 11.3.107), and the tenth/eleventh dorsal (MSF 11.3.241 and MSF 11.3.303). The MNI of juvenile Plateosaurus from bone field 11.3. is thus two.

Morphometric analysis

Neural arch size measured as zygapophyseal length

The values of zygapophyseal length of the isolated neural arches pertaining to juveniles and described here and of the specimens MSF 5B, MSF 23 and SMNS 13200 were measured for morphometric analysis (Table S1). The trend of zygapophyseal lengths along the cervical and dorsal series shows a clear pattern in all adult specimens studied (MSF 5B, MSF 23 and SMNS13200) (Fig. 16). This pattern is roughly followed by the disarticulated neural arches from bone field 11.3 as well. The anterior cervical neural arches show a rapid increase in zygapophyseal length, with C4/C5 showing the maximal length. Posteriorly, a decrease in the length of the cervical neural arches takes place, with the anterior dorsals (D3) showing the lowest value of zygapophyseal length. Subsequently the zygapophyseal length again increases, though at a much lower rate than in the anterior cervicals. The comparison of neural arches at the same positions suggests that the two juvenile individuals are of a slightly different size. The maximal size difference is approximately 20%.

Figure 16 Zygapophyseal lengths in the vertebral column of specimens MSF 11.3., MSF 5B, MSF 23 and SMNS 13200.

The zygapophyseal lengths of all specimens follow a distinct pattern throughout the vertebral column. The zygapophyseal lengths show a sharp increase in the anterior cervical series. The posterior cervicals decrease in length reaching their minimum length at the third dorsal neural arch. Afterwards they increase at a much lower rate than in the anterior cervical series. Specimen SMNS 13200 with the greatest femur length of all specimens studied, also shows greater zygapophyseal lengths. The juvenile MSF 11.3. specimens generally show a zygapophyseal length placed below of those from the mature specimens and only intervene with those of MSF 23 at some positions in the posteriormost cervical series.

Specimen SMNS 13200 with the greatest femur length (685 mm) generally possesses the greatest zygapophyseal lengths. Except for a few outliers, its lengths are clearly greater in comparison to the other specimens. Though specimen MSF 23 is the second largest individual on the basis of a femur length of 610 mm, the zygapophyseal lengths of the slightly smaller MSF 5B (calculated femur length of 565 mm), overlap with those of MSF 23. Throughout the vertebral series, the zygapophyseal lengths of the isolated neural arches are less than those of the adult specimens. The zygapophyseal lengths of the juveniles only overlap with those of specimen MSF 23 in the cervical series which may be due to the strong deformation in MSF 23.

Zyg/Fe ratios

Zygapophyseal length was calculated as a percentage of femur length (Table S2) to estimate femur length from the isolated neural arches (Table 3). With the help of these ratios, it is possible to estimate femur length of the juvenile specimens, which is documented in Table 3. Though the Zyg/Fe ratios of MSF 5B, MSF 23 and SMNS 13200 show a wide range between 12.5 and 28.3% (Table S2), they all reflect a pattern, following the regular change in zygapophyseal length throughout the vertebral column visible in all specimens. The pattern of increase and decrease of zygapophyseal lengths explains the wide range in the Zyg/Fe ratios in these individuals. The calculated femur lengths of the two 11.3. individuals range from 478.9 to 594.9 mm, depending on position of the neural arch and size of the individual. Again the variation in zygapophyseal length, which can be seen in all specimens studied, accounts for the relatively large variation in estimated femur length. Based on the vertebral positions that are represented twice, the femur length estimate for the larger juvenile is between 539 mm and 595 mm and that for the smaller juvenile is between 479 mm and 593 mm.

Table 3 Calculated range of femur length of the MSF 11.3. specimens.

For the calculation of a range of femur length of the juvenile MSF 11.3. specimens we only used the Zyg/Fe ratio of specimen SMNS 13200 due to completeness and good preservation of this specimen. The femur length of the juvenile specimens lies in between 479 and 595 mm. Lengths given in parentheses are again resulting from the longer specimen at positions occupied twice (refer to Table S1). The femur length estimate for the larger juvenile is between 539 mm and 595 mm and that for the smaller juvenile is between 479 mm and 593 mm.

Location	SMNS 13200
Zyg/Fe ratio (%)	MSF 11.3.
Zygapophyses
length (mm)	MSF 11.3.
Estimated femur
length (mm)	
C1				
C2 (axis)		77.4		
C3	21.4	117.7	549.2	
C4	25.0	142.5	570.0	
C5	25.1			
C6	22.1	129.7	586.3	
C7	25.8			
C8	25.5			
C9	19.6			
C10	19.0	109.9	578.4	
D1	17.8			
D2	16.5			
D3	16.1	77.1 (86.7)	478.9 (538.5)	
D4	16.6	98.7	593.15	
D5		94.2 (101.1)		
D6	19.7	108.5 (109.2)	550.5 (554.0)	
D7	20.4	106.6	521.5	
D8				
D9	17.8			
D10	20.4	110.8 (121.6)	542.1 (594.9)	
D11	20.7			
D12	19.6			
D13	20.9			
D14	19.4			
D15				

Discussion

Ontogenetic changes in vertebral morphology

Morphological changes through ontogeny in sauropodomorphs are poorly known because juveniles are rarely found and are mainly represented by late juveniles to subadult specimens (Ikejiri, Tidwell & Trexler, 2005; Tidwell & Wilhite, 2005). Until now there are just four basal sauropodomorphs and two sauropods with embryos or very young specimens known: Massospondylus carinatus (Reisz et al., 2005; Reisz et al., 2012), Mussaurus patagonicus (Bonaparte & Vince, 1979; Otero & Pol, 2013), Lufengosaurus (Reisz et al., 2013), the basal sauropodomorph Yunnanosaurus robustus (Sekiya et al., 2013), a baby titanosauriform closely related to Brachiosaurus (Carballido et al., 2012), and Europasaurus (Sander et al., 2006; Marpmann et al., 2011; Carballido & Sander, 2013). The most detailed study of ontogenetic changes in vertebral morphology has been done on Europasaurus holgeri, with different ontogenetic stages being recognized and defined (Carballido & Sander, 2013). Though in most cases isolated bones and incomplete specimens of vertebral column remains exacerbate studies on morphological changes through ontogeny (Carpenter & McIntosh, 1994; Foster, 2005).

Based on neural arch morphology, Carballido & Sander (2013) recognized five morphological ontogenetic stages: early immature, middle immature, late immature and two stages of maturity. In the early and middle immature stage, laminae and/or fossae of a neural arch are not fully developed. In the late immature stage all morphological characters of adults are already present, but the neurocentral suture remains open. The ontogenetic stage of the juvenile MSF 11.3. specimens equals the late immature stage found in Europasaurus holgeri.

The comparison of the morphology of cervical and dorsal neural arches between the juvenile MSF 11.3. specimens and the mature Plateosaurus did not reveal any differences at all. Laminae as well as fossae are all well developed in all osteologically mature individuals as well as in the juvenile Plateosaurus of bone field 11.3. The only distinction which can be made are the fully open neurocentral sutures in the 11.3. juveniles and the fully closed and invisible neurocentral sutures in the mature individuals (MSF 5B, MSF 23 and SMNS 13200).

The series of ontogenetic changes in the neural arch morphology as detected for Tazoudasaurus (Allain & Aquesbi, 2008), the brachiosaurid SMA 0009 (Carballido et al., 2012), Phuwiangosaurus (Martin, 1994) and especially the camarasauromorph Europasaurus holgeri (Carballido & Sander, 2013) cannot be observed in Plateosaurus. While this may be due to the late immature stage of the juveniles from bone field 11.3., it may be a plesiomorphy of basal sauropodomorphs. Basal sauropodomorphs are more plesiomorphic in their neural arch morphology than more derived sauropods and may have been more plesiomorphic in having less ontogenetic change in vertebral morphology. The function of laminae in sauropodomorphs was in the structural support of the neck and trunk region (Osborn, 1899; McIntosh, 1989), but also evolved as a correlate of axial pneumaticity (Seeley, 1870; Wilson, 1999; Taylor & Wedel, 2013). Most probably laminae can be explained by both factors.

Size and ontogenetic stage in Plateosaurus

The fully open neurocentral sutures of the neural arches described in this study are a reliable indicator for immaturity (Brochu, 1996). However, the calculated femur length for both juvenile individuals ranges between 479 mm and 595 mm, indicating that these were not smaller than many mature individuals from the Frick Plateosaurus bonebed. Histologically mature animals from Frick and Trossingen studied in Sander & Klein (2005) display a femur length between 480 mm and 990 mm. The femur lengths of osteologically immature, as well as osteologically mature, specimens and histologically mature animals merge into one another (Fig. 17). Furthermore, comparing the osteologically mature specimen MSF 5B (femur length: 565 mm) with the juveniles we assume that the immature animals would have become larger than MSF 5B. Our justification is the bone histology of a newly discovered (2012) associated skeleton (MSF 12.3.) with open neurocentral sutures in the cervical and dorsal column. Femoral histology of this individual indicates that it was still growing rapidly (see histological criteria in Sander & Klein, 2005; Klein & Sander, 2007) and was far from final body size. Importantly, both the current study and Sander and Klein’s study in 2005 show no correlation between age and size. Developmental plasticity is not only observable in histology of Plateosaurus, but also corroborated by its morphology.

Figure 17 Size and maturity stage corrobating developmental plasticity.

The femur lengths of the juvenile specimens of bone field 11.3. (blue) have been combined with the femur lengths of osteologically mature specimens studied: MSF 5B (red/black), MSF 23 (green/black) and SMNS 13200 (gray/black); and the femur lengths of histologically mature specimens (black) from Sander & Klein (2005). The femur length range of the juveniles has been divided up into 10 mm intervals to make it more practical in the diagram. The column diagram clearly shows the juvenile specimens and mature specimens merging into one another. The striking outlier of the whole diagram is specimen IFG with a remarkably great femur length of 990 mm. Nevertheless the diagram illustrates poor correlation between age (maturity) and size. Developmental plasticity is supported by histology as well as morphology.

However, as discussed in the introduction, alternative explanations to developmental plasticity such as the presence of several Plateosaurus species represented at the locality of Frick cannot be excluded, and a combination of several hypotheses (developmental plasticity, different species, populations separated in time, and/or sexual dimorphism) still remain possible and cannot be tested without further detailed study of the material from the Plateosaurus bonebeds and the taphonomy of the bonebeds.

Patterns of neurocentral suture closure

The isolated neural arches from bone field 11.3. contribute little to our understanding of the pattern of neurocentral suture closure in Plateosaurus. Circumstantial evidence consists of the lack of isolated posterior dorsals and caudal arches compared to the large number of caudal vertebrae preserved in the bone field (Fig. 18). This is suggestive of suture closure beginning in the tail and the posterior dorsal region. Further we lacked most of posterior cervical neural arches (C7–C9) in our sample. Those, as well as posterior dorsals (D12–D15), may have had completely closed neurocentral sutures and, despite being present in bone field 11.3, were not assignable to because the only reliable character for immaturity in our specimens (open neurocentral sutures) is not present. This observation might suggest a pattern of suture closure spreading from more than one vertebral position.

Figure 18 Caudal vertebra MSF 11.3.348.

MSF 11.3.348 is one of the caudal vertebrae in left lateral view found in bone field 11.3. The only morphological characters being present are the pre- and postzygapophyses. The neurocentral suture is completely closed as indicated by the dashed line. The whole caudal is covered by tectonic cracks. The scale bar measures 1 cm.

Implications for taphonomic hypothesis

As noted, the taphonomic hypothesis for the origin of the Plateosaurus bonebeds of Central Europe proposed by Sander (1992) predicted a size threshold for animals below which animals did not become mired. According to Sander (1992), this would explain the lack of juveniles because of their small size. While the discovery of juveniles in the lowermost bone horizon seemingly contradicts the hypothesis of Sander (1992), this is not the case. The juvenile Plateosaurus individuals described in this study are as large or even larger than the smallest fully grown Plateosaurus present at Frick, upholding the view that a size threshold existed that kept animals smaller than a 5-m Plateosaurus from becoming mired in the mud traps. This conclusion was implicit in the work of Sander & Klein (2005) and Klein & Sander (2007), but it was not expressed because histological immaturity could not be properly correlated with skeletal immaturity because isolated neural arches were not known from Frick at the time.

Conclusions

This study focuses on the first remains of juveniles of the basal sauropodomorph Plateosaurus engelhardti in Frick, Switzerland. P. engelhardti can be found in over 40 localities in Central Europe (Sander, 1992). The juveniles studied come from the locality of Frick, one of three localities preserving abundant remains of Plateosaurus and sharing the same taphonomy. These localities were described as Plateosaurus bonebeds by Sander (1992). The juveniles were found in a bone field in the lowermost bone horizon in the Gruhalde clay quarry of the Tonwerke Keller AG, revealing a concentration of several juvenile and adult bones. The most interesting specimens were isolated neural arches, representing an MNI of two juveniles that slightly differed in size. The juvenility and osteological immaturity of the remains can reliably be linked to the lack of fusion of the neural arches to the centra (Brochu, 1996). The ventral surface of the pedicel reveals the characteristic zipper-like surface of the suture, but the morphology of the immature neural arches does not differ from the morphology of the osteologically mature specimens (MSF 5B, MSF 23 and SMNS 13200) studied for comparison. Thus, the juvenile specimens of P. engelhardti seem to represent late immature individuals. Patterns of abundance in the bone field hint at a suture closure pattern in Plateosaurus from posterior to anterior. However, a pattern of suture closure spreading from more than one vertebral position is possible.

Morphometric analysis based on the ratio of zygapophyseal length to femur length indicates the femur length of the juvenile specimens to have been between 479 and 595 mm. Thus these animals were larger than the smallest histologically fully grown individual with a femur length of 480 mm from Frick (Sander & Klein, 2005) and most probably would have become larger than another individual with a femur length of 565 mm. The morphometric analysis thus independently confirms the poor correlation between age and size in the finds from Frick assigned to P. engelhardti, most likely reflecting pronounced developmental plasticity of Plateosaurus (Sander & Klein, 2005). However, alternative explanations such as the presence of several Plateosaurus species at the locality Frick cannot be excluded, and a combination of several hypotheses (developmental plasticity, different species, populations separated in time, and/or sexual dimorphism) still remain possible and cannot be tested without further detailed study of the material from the Plateosaurus bonebeds and the taphonomy of the bonebeds. Our study also failed to falsify the taphonomic miring hypothesis of Sander (1992) explaining the origin of the Plateosaurus bonebeds. While juvenile, the newly described individuals are not smaller than some adults and above the size threshold for miring. Institutional abbreviations

MSF Sauriermuseum Frick, Frick, Canton Aargau, Switzerland

NAA Naturama, Aarau, Canton Aargau, Switzerland

SMA Sauriermuseum Aathal, Aathal, Canton Zurich, Switzerland

SMNS Staatliches Museum für Naturkunde, Stuttgart, Germany

Anatomical abbreviations

acdl anterior centrodiapophyseal lamina

acpl anterior centroparapophyseal lamina

c centrum

Cd? caudal of indeterminate position

cdf centrodiapophyseal fossa

cpol centropostzygapophyseal lamina

cprl centroprezygapophyseal lamina

C1 atlas

C2 axis

C3 third cervical

C4 fourth cervical

C6 sixth cervical

C7 seventh cervical

C8 eighth cevival

C10 tenth cervical

C? cervical of indeterminate position

d diapophysis

D3 third dorsal

D4 fourth dorsal

D5 fifth dorsal

D6 sixth dorsal

D7 seventh dorsal

D10 tenth dorsal

D11 eleventh dorsal

D15 fifteenth dorsal

D? dorsal of indeterminate position

hypa hypantrum

hypo hyposphene

pa parapophysis

pcdl posterior centrodiapophyseal lamina

pocdf postzygapophyseal centrodiapophyseal fossa

podl postzygodiapophyseal lamina

poz postzygapophysis

ppdl paradiapophyseal lamina

prcdf prezygapophyseal centrodiapophyseal fossa

prdl prezygodiapophyseal lamina

prz prezygapophysis

spol spinopostzygapophyseal lamina

sprl spinoprezygapophyseal lamina

tpol intrapostzygapophyseal lamina

tprl intraprezygapophyseal lamina

Supplemental Information

Table S1 Zygapophyses lengths of the juvenile specimens MSF 11.3., MSF 5B, MSF 23 and SMNS 13200

Zygapophyses lengths in mm of the juvenile specimens MSF 11.3., MSF 5B, MSF 23 and SMNS 13200 with their respective position in the vertebral column. As noted in the description some positions can be recognized twice. The zygapophyses lengths in parentheses belong to the longer neural arch of those to be found twice in the sample. The specimen numbers of the 11.3. specimens is listed in Table 1. D3 (MSF 11.3.376/77.1 mm and MSF 11.3.360/86.7 mm). D5 (MSF 11.3.167/94.2 mm and MSF 11.3.067/101.1 mm). D6 (MSF 11.3.095/108.5 mm and MSF 11.3.107/109.2 mm). D10/D11 (MSF 11.3.241/110.8 mm and MSF 11.3.303/121.6 mm). Gaps are left, where data is missing due to preservation or the accessibility is was not given.

Click here for additional data file.

Table S2 Zyg/Fe ratios of MSF 5B, MSF 23 and SMNS 13200

Table S2 shows the calculated ratio in percent between the zygapophyses length and the femur length of the referred specimens. The ratios within on specimen and compared to each other specimen follow a determined scheme of zygapohyseal length explaining the wide range between 12.5 and 28.3%. With the help of the ratios we were able to define a new proxy for the determination of a range of femur lengths for the juvenile MSF 11.3. specimens.

Click here for additional data file.

Figure S1 Foil plan of bone field 11.3

The foil plan shows the isolated neural arches found in bone field 11.3. The neural arches are colored in green. The plan clearly shows that the neural arches were distributed over the whole area with no recognizable connection to each other and no centra lying next to them. The yardstick measures 1 m.

Click here for additional data file.

Figure S2 Specimen MSF 5 on exhibition in the SMA

Specimen MSF 5B reveals a complete articulated cervical series from vertebrae C2 to C10 and articulated dorsal vertebrae from D1 to D5. MSF 5B. All zygapophyseal lengths were available for measurements. MSF 5B being an osteologically mature specimen of Plateosaurus engelhardti shows completely closed neurocentral sutures with all morphological characters being well developed. Scale bar measures 5 cm.

Click here for additional data file.

Figure S3 Specimen MSF 23 on exhibition in the MSF

Specimen MSF 23 is a nearly complete and in most parts articulated P. engelhardti. The cervical vertebrae series is complete from C2 to C10. The dorsal series is complete from D1 to D15. The vertebrae of this specimen are heavily deformed, especially in the posterior dorsal series making measurements difficult. This specimen shows completely closed neurocentral sutures and all morphological characters are well developed. Scale bar measures 20 cm.

Click here for additional data file.

Figure S4 Specimen SMNS 13200: a cast exhibited in the NAA

A complete mounted skeleton cast of SMNS 13200 from Trossingen, Germany. The cervical as well as the dorsal vertebrae series is well preserved. All neurocentral sutures are completely closed and all morphological characters are well developed. For scaling: the left femur length of specimen SMNS 13200 measures 68.5 cm.

Click here for additional data file.

File S1 Three-dimensional model of specimen MSF 11.3.317 (axis)

Plateosaurus engelhardti from Frick, Switzerland. Three-dimensional model of an anterior neural arch of a late juvenile. Specimen MSF 11.3.317 shows prezygapophyses facets being smaller and shorter than those of the postzygapophyses. The spof is the only fossa developed. Note the zipper-like structures on the pedicels in ventral view. For scale bar refer to Fig. 2.

Click here for additional data file.

File S2 Three-dimensional model of specimen MSF 11.3.258 (C3)

Plateosaurus engelhardti from Frick, Switzerland. Three-dimensional model of an anterior neural arch of a late juvenile. The spof in MSF 11.3.258 gets deeper and the sprf developed. Note the zipper-like structures on the pedicels in ventral view. Furthermore the specimen shows tectonic cracks in dorsal view allover on the neural arch. For scale bar refer to Fig. 2.

Click here for additional data file.

File S3 Three-dimensional model of specimen MSF 11.3.339 (D7)

Plateosaurus engelhardti from Frick, Switzerland. Three-dimensional model of a middle dorsal neural arch of a late juvenile. The prcdf is extremely diminished in comparison to anterior dorsal neural arches. The articular surfaces of the prezygapophyses, postzygapophyses, and diapophyses display very rough articular surfaces once being covered by cartilage. Rough surfaces are an indicator of osteological immaturity. Specimen MSF 11.3.339 shows tectonic cracks in ventral view on the left lateral prezygapophysis. For scale bar refer to Fig. 13.

Click here for additional data file.

File S4 Three-dimensional model of specimen MSF 11.3.303 (D10/D11)

Plateosaurus engelhardti from Frick, Switzerland. Three-dimensional model of a middle/posterior dorsal neural arch of a late juvenile. In ventral view a partly preserved posterior caudal vertebrae (MSF 11.3.304) is cemented to specimen MSF 11.3.303. For scale bar refer to Fig. 15.

Click here for additional data file.

We want to thank Dr. Benedikt Pabst (Sauriermuseum Aathal and Sauriermuseum Frick, Switzerland) for the loan and entrustment of the juvenile specimens, giving us the opportunity to do research on the first juveniles of P. engelhardti to be found in Switzerland. Our gratitude also goes to Dr. Rainer Foelix (Naturama in Aarau, Switzerland) and Monica Rümbeli (Sauriermuseum Frick, Switzerland) for enabling access to the specimens SMNS 13200 and MSF 23 for further study. We thank Olaf Dülfer and Martin Schilling for preparing the material, as well as Georg Oleschinski for the photographs of the juvenile specimens. Our gratitude also goes to Franziska Sumpf for 3D-modelling four of the specimens studied. All four are employed at the Steinmann Institute of the University of Bonn. We want to thank Armin Schmitt (Dinosaurierpark Münchehagen) for the discussions and ideas on the morphometric part of this study. We also thank Adam Yates and Alejandro Otero for reviewing and improving our manuscript, and Jérémy Anquetin for editing. This research was performed as an MSc. Thesis at the University of Bonn.

Additional Information and Declarations

Competing Interests

Author Contributions

Neither author declares any competing interests regarding financial, non-financial, professional, or personal relationships that could in any way bias this research.

Rebecca Hofmann conceived and designed the experiments, performed the experiments, analyzed the data, wrote the paper, prepared figures and/or tables, reviewed drafts of the paper.

P. Martin Sander conceived and designed the experiments, analyzed the data, contributed reagents/materials/analysis tools, wrote the paper, prepared figures and/or tables, reviewed drafts of the paper.

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
