# Peer review of "The first juvenile specimens of Plateosaurus engelhardti from Frick, Switzerland: isolated neural arches and their implications for developmental plasticity in a basal sauropodomorph"

_PeerJ, doi:10.7717/peerj.458_

## Round 0.1 · original submission · Minor Revisions

Both reviewers provided detailed and constructive reviews of your manuscript. Their comments are reasonable and you should incorporate them into the revised version of your paper. Particularly, I think the following comments should be addressed in priority:

Reviewer 1:
- Incorporate the figures of the juvenile neural arches in the main paper. Current figures illustrating complete skeletons (Figs 1, 2, 3) can be sent to supplementary material.
- Consider the other option that the closure of the neurocentral sutures occurs when adult size is reached or nearly reached. This is the first thing that popped up into my non-specialist head when I first read your paper, so please discuss this in your revised version.

Reviewer 2:
- Provide more details regarding your assignment of these juvenile specimens to Plateosaurus engelhardti.

Please note that Reviewer 2 provided an annotated version of your manuscript. Be sure to check this thoroughly, including figure captions. Although PeerJ does not enforce strict formatting rules for the Reference section, please try to quickly short your references by Author, Year, Title, as suggested by Reviewer 2.

Additional comments from the Editor:
- I agree with Reviewer 2 that a map of Plateosaurus localities would be a great addition to the paper. I encourage you to add such a figure to your revised manuscript.
- You should slightly rephrase the Objectives of the study. You do not strictly ‘test the hypothesis’ of developmental plasticity. Developmental plasticity is rather one of the explanations (see comment from Reviewer 1) for your results. Similarly, you do not ‘test the miring hypothesis’ of Plateosaurus bonebed origins. As you say later in the manuscript: ‘we evaluate the implications of the finds of juveniles for the miring hypothesis’. You may want to check the abstract and introduction for similar overstatements.
- In order to improve readability, I would suggest splitting all of the items from Introduction to Anatomical abbreviations into two primary headings: Introduction and Material and Methods; with your original headings becoming secondary headings. You may also want to consider moving the anatomical abbreviations just after the Institutional abbreviations.
- You may also want to add a statement somewhere about the limited sample you use for your morphometric analysis. Three specimens, including one for which the femur is actually missing, is not a very large sample.
- Could Figure 4 be improved to look more ‘professional’? Follow the style of figures provided as supplementary material.
- Line 13: 'at the Heroldsberg' should be 'at Heroldsberg'
- Line 41: 'Geological setting' not 'Geologic setting'
- Line 44: 'geological' not 'geologic'
- Line 102: 'onmire'?
- Lines 117, 169, 546 (twice): 'osteologically' not 'osteological'
- Line 159: 'sometimes' not 'sometime'
- Lines 544, 546: 'histologically' not 'histological'

·

Basic reporting

The article is clear and on the whole well-written. There are however a few typos which I've listed in my general comments below.
I also wonder why the figures illustrating the juvenile neural arches (which are central to the whole paper) are treated as 'supplementary figures' while the far less important figures of the whole adult skeletons are included in the main paper. Consequently I had to download these sepparately and presumably that would be the case with the final publication. Given that peerJ is an electronic publication, there are no limits on space and it makes no sense to sepparate out these figures (except for the foil map which may be too large to include in the main pdf) so please include them in the main paper.

Experimental design

No comments

Validity of the findings

Generally I agree that the results support the conclusions however I think that the authors fail to emphasize the possibility that closure of the neurocentral sutures could have occurred particularly late in ontogeny, when adult size had been reached, or nearly reached. This reduces the amount of developmental plasticity that can be inferred. In particular I think the statement that the immature individuals would have reached a size greater MSF 5B is unsupported. It is entirely possible that growth in these late stage juveniles was already slowing drammatically and that size did not increase all that much while the neurocentral sutures closed.

Additional comments

There are a few minor points that need revising but do not affect the results or conclusions of this paper:

1. Juvenile Plateosaurus has been found before. The MCZ at Harvard have a nice juvenile Plateosaurus skull from the Fleming Fjord Formation of Greenland. However I do not know if this skull was illustatrated and /or reported in the account of the fauna:

Jenkins, FA et al. 1994. Late Triassic Continental Vertebrates and Depositional Environments of the Fleming Fjord Formation, Jameson Land, East Greenland (Meddelelser om Grønland - Geoscience No. 32)

I do not have access to a copy of this paper at the moment. The authors should check to determine wether or not the juvenile was reported before making the claim that the remains reported in this paper are the first juvenile Plateosaurus remains to be reported.

2. On line 31 Yates 2003 is cited as the authority for Plateosaurus gracilis. However I did not describe the species (Huene 1908 did that), I merely re-enstated the species as a member of the genus Plateosaurus (repeating an action already taken by Huene in 1926). Additionally the Yates 2003 paper cited in the references is the wrong one. It should be:

Yates AM. 2003. The species taxonomy of the sauropodomorph dinosaurs from the Löwenstein Formation (Norian, Late Triassic) of Germany. Palaeontology 46(2): 317-337.

3. On line 35, Prieto-Marquez and Norell (2011) should be cited as they have also weighed into the Plateosaurus species debate and ressurect the old name Plateosaurus erlenbergiensis (not that I agree that it is distinct from P. engelhardti).

4. The phrase 'ossification centre' as used (on lines 569 and 595-596)is potentially confusing as it is generally used to indicate a point or region within an individual bone around which ossification occurs during development. I would rewrite the sentence on 569 along the lines: "This indicates a pattern of suture closure spreading from more than one vertebral position." The sentence on 595-596 also needs to be rewritten for clarity.

5. Lastly there are a few typos that need correcting (there are probably more that I didn't catch)

Line 8. Sauropodomorpha needs to begin with a capital letter.
Line 154. Should read "how many animals are represented"
Line 156. Should read "two juvenile animals"
Line 302. should read "not preserved"

·

Basic reporting

"No Comments"

Experimental design

"No Comments"

Validity of the findings

"No Comments"

Additional comments

The manuscript of Hoffman & Sander presents the description of juvenile specimens of the basal sauropodomorph Plateosaurus engelthardti, from the locality of Frick, Switzerland, and attempting to add support to the current hypothesis of developmental plasticity in this species and also the taphonomic scenario surrounding the bonebed this specimens were found.
The manuscript is well written, clear and concise, and the descriptions are accurate.
I find this manuscript interesting because juvenile specimens of basal sauropodomorphs are scarce. To date, embryos or juvenile specimens of basal sauropodomorphs are known only from few taxa (Mussaurus, Massospondylus, Lufengosaurus, and Yunnanosaurus). Hence any new study of basal sauropodomorph juvenile specimens adds valuable information. Besides, if the taxonomic validity of juveniles presented in this contribution is correct, this is the first report of P. engelthardti juveniles.
Considering all the above exposed, I recommend the publication of this manuscript in PeerJ after a minor revision.

I found, however, some minor issues that the authors must consider before publication in PeerJ.

As the authors pointed out, a key issue surrounding hypothesis of developmental plasticity is that the sample derived from the single species. There is consensus that the identity of the Frick specimens belong to a single species (regardless of the nomenclatural controversies) with some variations related to intraspecifc variation, sex dimorfism, etc. (e.g., Klein and Sander, 2007). Nonetheless, as the material described herein has not been published before (as far as I know), it would be reasonable that the authors devote some lines to justify the assignment of the material studied to Plateosaurus engelthardti and not just saying: “it belongs to P. engelthardti because all the material from Frick was previously assigned to P. engelthardti”. What I mean with this is: if you notice similarities between this new material and the previously published material from Frick, just pointed out in the paper. This is of parmount importance when attempting to test/reinforce such interesting hypothesis like the developmental plasticity. To solve this, you can either add a new heading like “Assignment of juveniles to P. engelthardti” (or something like that), or just make some comment within the descriptions when necessary. This will give more support to the developmental plasticity hypothesis.
Apart from this, I noticed no comparisons with other taxa in the descriptions. Again, pointing out differences (if present) with other taxa, or with other Plateosaurus species will give more support regarding the assignment of the juveniles to P. engelthardti, and hence, more support to the developmental plasticity hypothesis.

The remaining comments are inserted in the PDF file.

I think there will be not necessary another round of review, but, if necessary, I will be glad to make a final review.

---

## Round 0.2 · Minor Revisions

You have done a great job in addressing the reviewers comments. I think the addition of the 3D models is a very good idea.

Before accepting your paper I would like you to quickly address the following important issue: you cite two publications by Yates (2003) without differentiating them. Please, go through your manuscript one last time and differentiate them as Yates (2003a) and Yates (2003b), without forgetting the reference list.

While you are at it, please also address the following:
- Line 3 in the Abstract: 'sauropodomorph', not 'sauropdomorph'
- Line 130: a space is missing before 'Chinsamy-Turan, 2005'
- Lines 374–379: add a note saying that these 3D models are provided as supplemental files 1—4
- Lines 771–783: please, consider thanking Adam Yates and Alejandro Otero for their comments.

You should be able to address these minor issues fairly quickly, then I will formally accept your manuscript for publication in PeerJ.

---

## Round 0.3 · accepted · Accept

I am glad you were able to address these last few modifications so quickly. Your paper is now accepted for publication, congratulations!